# Ethical challenges of using remote monitoring technologies for clinical research: A case study of the role of local research ethics committees in the RADAR-AD study

Marijn Muurling[1,2]*, Anna M. G. Pasmooij[3], Ivan Koychev[4], Dora Roik[5], Lutz Froelich[5], Emilia Schwertner[6,7], Dorota Religa[6,8], Carla Abdelnour[9], Mercè Boada[10,11], Monica Almici[12], Samantha Galluzzi[12], Sandra Cardoso[13], Alexandre de Mendonça[13], Andrew P. Owens[14], Sajini Kuruppu[14], Martha Therese Gjestsen[15], Ioulietta Lazarou[16], Mara Gkioka[17], Magda Tsolaki[17], Ana Diaz[18], Dianne Gove[18], Pieter Jelle Visser[1,2,6,19], Dag Aarsland[14,15], Federica Lucivero[20☯], Casper de Boer[1,2☯], the RADAR-AD Consortium[¶]

1 Alzheimer Center Amsterdam, Neurology, Vrije Universiteit Amsterdam, Amsterdam UMC location VUmc, Amsterdam, The Netherlands, 2 Amsterdam Neuroscience, Neurodegeneration, Amsterdam, The Netherlands, 3 Dutch Medicines Evaluation Board, Utrecht, Netherlands, 4 Department of Psychiatry, University of Oxford, Oxford, United Kingdom, 5 Department of Geriatric Psychiatry, Central Institute for Mental Health, Medical Faculty Mannheim, University of Heidelberg, Heidelberg, Germany, 6 Division of Clinical Geriatrics, Department of Neurobiology, Department of Care Sciences and Society, Karolinska Institutet, Stockholm, Sweden, 7 Faculty of Psychology, SWPS University of Social Sciences and Humanities, Krakow, Poland, 8 Theme Inflammation and Aging, Karolinska University Hospital, Huddinge, Sweden, 9 Department of Neurology and Neurological Sciences, Stanford University School of Medicine, Stanford, CA, United States of America, 10 Ace Alzheimer Center Barcelona–Universitat Internacional de Catalunya, Barcelona, Spain, 11 Networking Research Center on Neurodegenerative Diseases (CIBERNED), Instituto de Salud Carlos III, Madrid, Spain, 12 Laboratory Alzheimer's Neuroimaging & Epidemiology, IRCCS Istituto Centro San Giovanni di Dio Fatebenefratelli, Brescia, Italy, 13 Faculdade de Medicina da Universidade de Lisboa, Lisbon, Portugal, 14 Institute of Psychiatry, Psychology and Neuroscience, King's College London, London, United Kingdom, 15 Centre for Age-Related Medicine, Stavanger University Hospital, Stavanger, Norway, 16 Information Technologies Institute, Center for Research and Technology Hellas (CERTH-ITI), Thessaloniki, Greece, 17 Laboratory of Neurodegenerative Diseases, Center for Interdisciplinary Research and Innovation (CIRI–AUTh), Balkan Center, Aristotle University of Thessaloniki, Thessaloniki, Greece, 18 Alzheimer Europe, Luxembourg, Luxembourg, 19 Department of Psychiatry and Neuropsychology, School for Mental Health and Neuroscience, Maastricht University, Maastricht, The Netherlands, 20 Ethox and Wellcome Centre for Ethics and Humanities, University of Oxford, Oxford, United Kingdom

☯ These authors contributed equally to this work.
¶ Membership of the RADAR-AD Consortium is listed in the Acknowledgments.
* m.muurling@amsterdamumc.nl

**Data Availability Statement:** The data are confidential documents from the local ethical committees. We can share the data upon

## Abstract

### Introduction

Clinical research with remote monitoring technologies (RMTs) has multiple advantages over standard paper-pencil tests, but also raises several ethical concerns. While several studies have addressed the issue of governance of big data in clinical research from the legal or ethical perspectives, the viewpoint of local research ethics committee (REC) members is underrepresented in the current literature. The aim of this study is therefore to find which specific ethical challenges are raised by RECs in the context of a large European study on remote monitoring in all syndromic stages of Alzheimer's disease, and what gaps remain.

reasonable request via radar.ad@amsterdamumc.nl.

**Funding:** The RADAR-AD project has received funding from the Innovative Medicines Initiative 2 Joint Undertaking under grant agreement No 806999. This Joint Undertaking receives support from the European Union's Horizon 2020 research and innovation programme and EFPIA and Software AG. See https://www.imi.europa.eu/ for more details. This communication reflects the views of the RADAR-AD consortium and neither IMI nor the European Union and EFPIA are liable for any use that may be made of the information contained herein. Research of Alzheimer center Amsterdam is part of the neurodegeneration research program of Amsterdam Neuroscience. Alzheimer Center Amsterdam is supported by Stichting Alzheimer Nederland and Stichting Steun Alzheimercentrum Amsterdam. IK declares support for this work through the National Institute of Health Research (personal award and Oxford Health Biomedical Research Centre) and the Medical Research Council (Dementias Platform UK grant). CA's postdoctoral fellowship is funded by the Susan and Charles Berghoff Foundation. The funders had no role in study design, data collection and analysis, decision to publish, or preparation of the manuscript.

**Competing interests:** The authors have read the journal's policy and have the following competing interests: IK is a paid medical advisor for digital healthcare technology companies Five Lives SAS and Cognetivity Ltd., outside the submitted work. CA has received honoraria as speaker from F. Hoffmann-La Roche Ltd, Zambon, Nutricia, Schwabe Farma Ibérica S.A.U, outside of the submitted work. CA is a member of the Board of Directors of the Lewy Body Dementia Association, outside the submitted work. DA has received research support and/or honoraria from Astra-Zeneca, H. Lundbeck, Novartis Pharmaceuticals, Biogen, and GE Health, and served as paid consultant for H. Lundbeck, Eisai, Heptares, Mentis Cura, and Roche Diagnostics, outside the submitted work. MB is an employee of the Ace Alzheimer Center and an advisory board member for Grifols, Roche, Eli Lilly, Araclon Biotech, Merck, Zambon, Biogen, Novo Nordisk, Bioiberica, Eisai, Servier, and Schwabe Pharma, outside the submitted work. This does not alter our adherence to PLOS ONE policies on sharing data and materials. All other authors have declared that no competing interests exist. There are no patents, products in development or marketed products associated with this research to declare.

## Methods

Documents describing the REC review process at 10 sites in 9 European countries from the project Remote Assessment of Disease and Relapse–Alzheimer's Disease (RADAR-AD) were collected and translated. Main themes emerging in the documents were identified using a qualitative analysis approach.

## Results

Four main themes emerged after analysis: data management, participant's wellbeing, methodological issues, and the issue of defining the regulatory category of RMTs. Review processes differed across sites: process duration varied from 71 to 423 days, some RECs did not raise any issues, whereas others raised up to 35 concerns, and the approval of a data protection officer was needed in half of the sites.

## Discussion

The differences in the ethics review process of the same study protocol across different local settings suggest that a multi-site study would benefit from a harmonization in research ethics governance processes. More specifically, some best practices could be included in ethical reviews across institutional and national contexts, such as the opinion of an institutional data protection officer, patient advisory board reviews of the protocol and plans for how ethical reflection is embedded within the study.

## Introduction

Digital devices, data analytics and artificial intelligence are rapidly changing the way clinical research is conducted, as a variety of aspects of people's health and lifestyle can be collected in real-time and patterns in large datasets can be identified. Remote monitoring technologies (RMTs), such as wearables, smartphone applications, and fixed sensors at home can capture real-world information about study participants in a continuous and objective manner. RMTs therefore have several potential advantages over in-clinic pen-and-paper tests that are usually assessed periodically, rely on the participant's or partner's recall and can only be completed in a controlled environment of the clinic [1]. Such advantages of RMTs are especially helpful in health conditions with a long preclinical phase, such as Alzheimer's disease (AD) [2, 3], or recurring episodes, such as depression [4]. However, while RMTs open new opportunities for clinical research, the use of digital technologies also raises new questions for research ethics, including obvious concerns regarding privacy and data security [5, 6]. Moreover, although participants consent to participate in the study, it is almost impossible to inform participants completely, simply and clearly about what types of data are being collected, who can access the data and how the data will be analyzed [7, 8]. Another important concern is equality, meaning that people who do not own a smartphone–whether for economic, educational or geographical reasons–will have limited possibilities to participate in RMT research with consequences for reliability and generalizability [9].

While several studies have addressed the issue of governance of big data in clinical research from the legal perspective [10–12] or ethical perspective [13–16], the perspective of institutional research ethics committee (REC) members is underrepresented in the current body of

literature [17]. Research ethics committees (RECs) enforce research governance through a review mechanism, reviewing the study design and protocols, inclusion and exclusion criteria for participants, informed consent procedures and data safety, management and monitoring plans. The composition of RECs is diverse, so that a comprehensive set of perspectives is considered. Variation in how research governance practices are enforced across different institutions and countries participating in multi-site studies is a concern. With the introduction of novel technologies for data collection and analysis, such as RMTs, the role of the RECs becomes central in assessing the legitimate, fair, and ethical use of those technologies in research settings. To this end, we must better understand the 'needs, views and attitudes' of REC members wherever health-related RMTs are used as a source of 'big data' in research [17]. This study addresses this knowledge gap by studying the views of REC members in the context of a large pan-European study on remote monitoring in AD [2], using a qualitative analysis approach. This paper explores the type of issues that are currently highlighted by RECs, how they are addressed within a project, what gaps remain and provide suggestions on how these gaps may be addressed. We describe the specific challenges raised by RECs using examples from the European research project Remote Assessment of Disease and Relapse–Alzheimer's Disease (RADAR-AD) [2], which aims to validate how RMTs assess functional decline in AD. Since the RADAR-AD study protocol is implemented in 10 European countries, this material gives us an opportunity to compare ethical questions across European countries.

## Materials and methods

The RADAR-AD study (www.radar-ad.org) is a European multi-center observational study, aiming to find and validate remote monitoring technologies that measure functional and cognitive decline in AD [2]. This study included over 220 participants, equally distributed in four study groups: healthy controls, preclinical AD, prodromal AD, and mild-to-moderate AD participants. Participants visited the clinic at the start of the data collection period and were thereafter monitored in the real-world for eight weeks, using the RMTs presented in Table 1. RADAR-AD identified these RMTs based on an elaborate literature study and asking patients [18] and only used RMTs that were already publicly available. No data safety monitoring

**Table 1. Remote monitoring technologies in the RADAR-AD protocol.**

| RMT | Description | Usage | Assesses |
|---|---|---|---|
| **Fitbit charge 3** | A wrist-worn activity tracker | At home, measures continuously for 8 weeks | Heart rate, step count and sleep patterns |
| **Axivity AX3** | A wrist-worn activity tracker [19] | At home, measures continuously for 8 weeks | Activity and sleep patterns |
| **RADAR passive RMT app** | An Android-based smartphone application that passively collects data on phone usage [20] | At home, measures continuously for 8 weeks | Phone usage, surroundings, location |
| **Autographer wearable camera** | A camera which is worn around the neck and takes a photo every 20 seconds. The use of the camera is optional [21] | At home, three times 2 consecutive days (self-chosen by the participant) | Activities of daily living |
| **Mezurio app** | A smartphone application presenting daily cognitive tasks and short questionnaires [22, 23] | At home, twice per day for 8 weeks | Planning skills, memory, keyboard dynamics, language sleep quality and mood |
| **Altoida app** | A smartphone and tablet based application that simulates a complex activity of daily living using augmented reality [24] | Once at the clinic and weekly at home | Spatial navigation, memory, functional impairment, and motor functions |
| **Gait up Physilog sensors** | Three body-worn sensors containing accelerometers and gyroscopes during three short walk tests [25] | Once in the clinic | Gait |
| **Banking app** | An application simulating a bank withdrawal [26] | Once in the clinic | Functional abilities of managing finances |

board (DSMB) was installed during the study, as the risk assessment of the sponsor (Amsterdam UMC) showed that overall risks were low.

## Data collection

Documents describing the REC review process at ten RADAR-AD study sites were collected for analysis. These sites included clinical academic sites specializing in brain health and dementia in Amsterdam (The Netherlands), Barcelona (Spain), Brescia (Italy), Lisbon (Portugal), London and Oxford (United Kingdom), Mannheim (Germany), Stavanger (Norway), Stockholm (Sweden) and Thessaloniki (Greece) (Table 2). RECs operated either on a site-specific, region-specific, or country-specific level (Table 2), meaning that that particular REC reviews studies from that site only, from several sites in that region, or from sites in the entire country, respectively. Amsterdam UMC (Amsterdam site) was the sponsor of the study and therefore coordinated the clinical study. The other RADAR-AD sites (see Muurling, de Boer [2]) had not yet obtained ethical approval yet by the time of writing, mainly due to legal contract issues, and were therefore excluded from the current study. The REC process for King's College London and University of Oxford was centralized and was therefore analyzed as one. These documents included, but were not limited to, primary REC submission documents, feedback on the submission from RECs, response letters to feedback from RECs, and correspondence between local study teams and RECs, Data Protection Officers (DPO's), and scientific review boards (More detailed information in S1 Table). Documents were anonymized and translated into English by local study teams. At each RADAR-AD study site, written permission was obtained from the REC to use direct anonymized quotes for publication.

## Data analysis and paper drafting

The translated REC documents were imported into Atlas.ti version 9 [27], a software package for qualitative analysis of research documents. Following thematic analysis methods [28],

**Table 2. The participating sites and the REC that approved the study.**

| Country | City | Institution | REC | Level |
|---------|------|-------------|-----|-------|
| **Germany** | Mannheim | Zentralinstitut für Seelische Gesundheit Mannheim | Ethics Committee II of the Ruprecht-Karls-University of Heidelberg (Medical Faculty Mannheim) | Site-specific |
| **Greece** | Thessaloniki | Aristotle University of Thessaloniki | Ethics Committee of Medical Faculty of Aristotle University of Thessaloniki and Ethics Committee of Alzheimer Hellas | Site-specific |
| **Italy** | Brescia | IRCCS Istituto Centro San Giovanni di Dio Fatebenefratelli | Comitato Etico IRCCS Centro San Giovanni di Dio–Fatebenefratelli di Brescia | Site-specific |
| **Norway** | Stavanger | Centre for Age-Related Medicine | Regionale komiteer for medisinsk og helsefaglig forskningsetikk | Region-specific |
| **Portugal** | Lisbon | Faculdade de Medicina da Universidade de Lisboa | Comissão de Ética do Centro Académico de Medicina de Lisboa | Site-specific |
| **Spain** | Barcelona | Ace Alzheimer Center Barcelona | Drug Research Ethics Committee (CEIm) of Universitat International de Catalunya | Site-specific |
| **Sweden** | Stockholm | Karolinska Institutet | Swedish Ethical Review Authority | Country-specific |
| **The Netherlands** | Amsterdam | Amsterdam UMC | Medisch Ethische Toetsingscommissie VUmc | Site-specific |
| **United Kingdom** | Oxford | University of Oxford | London–West London & GTAC (Gene Therapy Advisory Committee) Research Ethics Committee | Region-specific (REC gets randomly allocated across the country) |
|  | London | King's College London |  |  |

Note. Last column: RECs operated either on a site-specific, region-specific, or country-specific level, meaning that that particular REC reviews studies from that site only, from several sites in that region, or from sites in the entire country, respectively.

main authors FL, CdB and MM read a predefined subset of the documents independently and formulated a deductive codebook based on an internal discussion on emerging issues in the study documents and issues highlighted in existing literature. FL, CdB, and MM independently coded a subset of the study documents using the agreed codebook, while adding codes inductively when necessary [29]. New codes and emerging themes were discussed in subsequent meetings. At last, MM reviewed all coded documents to harmonize the coding.

The main findings were drafted based on the primary analysis by FL, CdB, and MM and submitted to all co-authors together with several discussion points. A larger meeting was organized to discuss the findings and a first draft of the discussion was drafted by FL, CdB and MM. This draft was circulated among co-authors who further contributed to it. All co-authors agreed to the final version of the manuscript.

## Results

The analysis of the documents highlighted the diversity of the REC review process as well as four emerging types of issues that we clustered in four categories: data management, participant's wellbeing, methodological issues, and definition of the devices. Code frequencies are available in Table 3, while code descriptions and exemplary quotes are listed in Tables 4–7.

Fig 1 describes the duration of the REC review process at 10 different RADAR-AD study locations. More detailed information on the review processes can be found in the S1 Table. At

**Table 3. Code frequencies per site.**

| | Totals | Centre for Age-Related Medicine (Stavanger) | | IRCCS Istituto Centro San Giovanni di Dio Fatebenefratelli (Brescia) | | Zentralinstitut für Seelische Gesundheit Mannheim (Mannheim) | | King's College London (London) / University of Oxford (Oxford) | | Karolinska Institutet (Stockholm) | | Ace Alzheimer Center Barcelona (Barcelona) | | Faculdade de Medicina da Universidade de Lisboa (Lisbon) | | Amsterdam UMC (Amsterdam) | | Aristotle University of Thessaloniki (Thessaloniki) | |
|---|---|---|---|---|---|---|---|---|---|---|---|---|---|---|---|---|---|---|---|
| | N | N | % | N | % | N | % | N | % | N | % | N | % | N | % | N | % | N | % |
| **Data management** | | | | | | | | | | | | | | | | | | | |
| Data access | 3 | 1 | 14.3% | 0 | 0.0% | 0 | 0.0% | 1 | 7.7% | 0 | 0.0% | 0 | 0.0% | 0 | 0.0% | 1 | 2.9% | 0 | 0.0% |
| Data protection | 16 | 0 | 0.0% | 2 | 50.0% | 9 | 39.1% | 0 | 0.0% | 0 | 0.0% | 1 | 20.0% | 0 | 0.0% | 4 | 11.4% | 0 | 0.0% |
| Data security | 10 | 1 | 14.3% | 1 | 25.0% | 1 | 4.3% | 1 | 7.7% | 3 | 12.5% | 0 | 0.0% | 0 | 0.0% | 3 | 8.6% | 0 | 0.0% |
| Data sharing | 6 | 1 | 14.3% | 0 | 0.0% | 2 | 8.7% | 3 | 23.1% | 0 | 0.0% | 0 | 0.0% | 0 | 0.0% | 0 | 0.0% | 0 | 0.0% |
| Data storage | 6 | 3 | 42.9% | 0 | 0.0% | 1 | 4.3% | 1 | 7.7% | 1 | 4.2% | 0 | 0.0% | 0 | 0.0% | 0 | 0.0% | 0 | 0.0% |
| **Participant's wellbeing** | | | | | | | | | | | | | | | | | | | |
| Comfort | 3 | 0 | 0.0% | 0 | 0.0% | 0 | 0.0% | 0 | 0.0% | 2 | 8.3% | 1 | 20.0% | 0 | 0.0% | 0 | 0.0% | 0 | 0.0% |
| Competence | 4 | 0 | 0.0% | 0 | 0.0% | 1 | 4.3% | 1 | 7.7% | 0 | 0.0% | 1 | 20.0% | 0 | 0.0% | 1 | 2.9% | 0 | 0.0% |
| Incidental findings | 4 | 0 | 0.0% | 0 | 0.0% | 0 | 0.0% | 2 | 15.4% | 1 | 4.2% | 0 | 0.0% | 0 | 0.0% | 1 | 2.9% | 0 | 0.0% |
| Privacy | 14 | 1 | 14.3% | 1 | 25.0% | 4 | 17.4% | 1 | 7.7% | 5 | 20.8% | 0 | 0.0% | 0 | 0.0% | 2 | 5.7% | 0 | 0.0% |
| Proportionality | 7 | 0 | 0.0% | 0 | 0.0% | 0 | 0.0% | 0 | 0.0% | 4 | 16.7% | 0 | 0.0% | 1 | 20.0% | 2 | 5.7% | 0 | 0.0% |
| Resistance | 1 | 0 | 0.0% | 0 | 0.0% | 0 | 0.0% | 0 | 0.0% | 0 | 0.0% | 0 | 0.0% | 0 | 0.0% | 1 | 2.9% | 0 | 0.0% |
| Safety | 5 | 0 | 0.0% | 0 | 0.0% | 1 | 4.3% | 0 | 0.0% | 4 | 16.7% | 0 | 0.0% | 0 | 0.0% | 0 | 0.0% | 0 | 0.0% |
| Stigmatisation | 0 | 0 | 0.0% | 0 | 0.0% | 0 | 0.0% | 0 | 0.0% | 0 | 0.0% | 0 | 0.0% | 0 | 0.0% | 0 | 0.0% | 0 | 0.0% |
| **Methodological issues** | | | | | | | | | | | | | | | | | | | |
| Liability | 4 | 0 | 0.0% | 0 | 0.0% | 1 | 4.3% | 0 | 0.0% | 0 | 0.0% | 0 | 0.0% | 1 | 20.0% | 2 | 5.7% | 0 | 0.0% |
| Methodological concerns | 21 | 0 | 0.0% | 0 | 0.0% | 1 | 4.3% | 2 | 15.4% | 3 | 12.5% | 1 | 20.0% | 2 | 40.0% | 12 | 34.3% | 0 | 0.0% |
| Recruitment | 5 | 0 | 0.0% | 0 | 0.0% | 0 | 0.0% | 1 | 7.7% | 1 | 4.2% | 0 | 0.0% | 1 | 20.0% | 2 | 5.7% | 0 | 0.0% |
| **Definition of the devices** | | | | | | | | | | | | | | | | | | | |
| Medical devices | 7 | 0 | 0.0% | 0 | 0.0% | 2 | 8.7% | 0 | 0.0% | 0 | 0.0% | 1 | 20.0% | 0 | 0.0% | 4 | 11.4% | 0 | 0.0% |
| Totals | 116 | 7 | 100% | 4 | 100% | 23 | 100% | 13 | 100% | 24 | 100% | 5 | 100% | 5 | 100% | 35 | 100% | 0 | 100% |

Number of times a theme was mentioned in the ethics review documents from that particular REC, as absolute number (n) and percentage per site (%), clustered on themes. Darker green indicates higher percentages.

Table 4. Issues raised about data management.

| Code | Description | Example quote | Number of sites |
|---|---|---|---|
| **Data access** | Describes issues related to who has access to the data during and within the study | 'How is privacy guaranteed? Support of IT personnel is possibly needed to install the app, downloading data from the RMTs and/or prevent malfunction, so that there is a chance that IT personnel sees sensitive information. Is a confidentially and/or processing agreement formalized?' | 3 |
| **Data protection** | Is used when explicit reference to General Data Protection Regulation (GDPR) or other data protection regulations is made | 'Provide an assessment of impact on the protection of data, which guarantees compliance with the GDPR and the additional provision XVII of the Organic Law 3/2018 of December 5, of personal data protection and guarantee of the digital rights that are collected from the smartphone application 'Mezurio', the photographs captured by the camera 'Autograph', the application 'RADAR-base' and the devices 'Fitbit charge 3' and 'Axivity AX3.' | 4 |
| **Data security** | Is used when there is an explicit reference to data security, in contrast to the code 'safety', which is used for participant's safety only. | 'Banking app: how is data secured? How is abuse and accusation of abuse prevented?' and 'It should be informed that participants agree to the data being stored at Fitbit. The information sheets should address what this entails. Here you should seek advice from the data protection officer.' | 6 |
| **Data sharing** | Describes issues regarding third parties using the data, i.e., parties that are not related to the project. | 'Make it clearer in the participant information sheets [...] that only anonymized data will be made available to RADAR-AD sites/organizations outside the confines of the study itself.' and 'If the planned data recipients were persons and companies based outside the EU (in particular the USA), the data transfer would only be permissible under the conditions of the EU GDPR.' | 3 |
| **Data storage** | Describes issues regarding storing data, for example sufficient storing capacity, storage duration and the storage place. | 'It must be stated that data will only be analyzed until the end of the project. After the end of the project, it is allowed to store data for 5 years, but only for follow-up.' and 'In the event of revocation, all data must be deleted if they are not anonymized.' | 4 |

Table 5. Issues raised about participant's wellbeing.

| Code | Description | Example quote | Number of sites |
|---|---|---|---|
| **Comfort** | Refers to the concern that participants could experience mental or physical discomfort during the study. | 'Neuropsychological tests can be tiring. The remote measurement techniques must be used every day, which can lead to discomfort [...].' | 2 |
| **Competence** | Describes issues regarding participant's (mental and physical) ability to participate in the study. | 'In this study, can all study participants be considered mentally competent?' | 5 |
| **Incidental findings** | Refers to issues related to finding unexpected (medical) outcomes during the study procedures. This code applies both to unexpected cognitive or medical findings and illegal activities. | 'What happens when a healthy control subject has similar scores on the study tests as an Alzheimer subject?', and 'if the images [of the wearable camera] show illegal activities, according to national law, the researcher may have a legal and professional obligation to breach confidentiality and pass on image data to appropriate authorities.' | 4 |
| **Privacy** | Describes concerns regarding the privacy of the participant and their environment during data collection, including privacy of partners or bystanders. | 'How are the researchers planning to deal with the privacy of bystanders whose images are being recorded while the study subject wears the camera?', 'How is the privacy and data-protection of study subjects safeguarded on the RADAR-base platform?', or 'Statements made by the caregiver about the patient may not be used and processed without the patient's consent.' | 6 |
| **Proportionality** | Is used to describe concerns about asking participants too much. | 'The study protocol states that study partners also have to install the RADAR-base app on their smartphone. Please justify why this is necessary.' | 3 |
| **Resistance** | Describes references to participants being suspicious or resisting the study. | 'How is patient confusion/suspicion prevented; how is resistance of the patient handled?' | 1 |
| **Safety** | Is used when there are concerns that a device or participation in the study could jeopardize participant's safety. This does not include data safety, but a participant's physical and mental wellbeing only. | '[...] wearing cameras in everyday situations can potentially be associated with a certain risk. [...] Possible scenarios include the following: the participant is questioned by (potentially hostile or suspicious) third parties objecting to unwanted image recording, thus posing a threat to the participant's safety.' | 2 |

**Table 6. Issues raised about methodology.**

| Code | Description | Example quote | Number of sites |
|---|---|---|---|
| Methodological concerns | Concerns related to study objectives, study design, outcome measures, data analyses (statistical analysis) or the choice of a certain RMT, without explicit reference to a specific ethical issue. | *'There is a high chance of incorrect or missing data. How will be dealt with this statistically?', 'Do control subjects also need a study partner? If yes, please explain why.', or 'Please provide a clear study objective, describing what is being sought in respect of each participant group in respect of assessing the feasibility of the remote technologies in each group.'.* | 6 |
| Recruitment | Describes issues related to recruitment of participants in the study. | *'Please provide a clear recruitment process of the three groups of participants, explaining how they will be identified, approached and consented into the study.'* | 4 |
| Liability | Refers to issues of responsibility or liability, related to the use of medical devices, data management, patient safety, or other. | *'Who has liability when a device gets lost or suffers damage while in the possession of a study subject?'* | 4 |

all study locations, the same research protocol was submitted to the REC. The duration of the REC process, i.e., from primary submission to obtaining approval, differed significantly across the consortium. The fastest approval was obtained in Faculdade de Medicina da Universidade de Lisboa (Lisbon site, 71 days), while the longest approval process took place in Karolinska Institutet (Stockholm site, 423 days). The average duration of the REC process was 232 (SD 145) days. Most study sites (5 out of 9) received one round of comments before REC approval was obtained. One study site (Aristotle University of Thessaloniki, Thessaloniki site) obtained direct approval after the primary submission, while Amsterdam UMC (Amsterdam site) and Karolinska Institutet (Stockholm site) had to go through 4 rounds to obtain approval. In 5 out of 9 study locations, the protocol also required approval from a Data Protection Officer (DPO) (S1 Table). In Amsterdam UMC (Amsterdam site), additional approval had to be obtained from an independent scientific board and the institution's Information Technology (IT) department. Also, in the Centre for Age-Related Medicine (Stavanger site), the study could not start until the institution's IT-department had approved the RMTs used.

All RECs were concerned about several issues related to **data management,** such as data access, data security, data sharing, data storage, and data protection (Table 4), except for the sites in Lisbon and Thessaloniki. These issues represented most of the expressed concerns. In Amsterdam UMC (Amsterdam site), and the Centre for Age-Related Medicine (Stavanger site), the security of the apps, devices and data platform had to be checked by a specialized IT security department. Data security was often intertwined with data sharing, data storage and data protection issues, for example: *'The REC requests that [Sites name]'s routines will be followed for secure data collection, data transfer and data storage, and that one contacts the data protection officer at [Sites name] for guidance.'* Regarding data protection, the REC suggested that most sites seek the advice of a DPO to check data protection. The use of the wearable camera during the RADAR-AD study was often mentioned in relation to the data protection issues. Faculdade de Medicina da Universidade de Lisboa (Lisbon site), and IRCCS Istituto

**Table 7. Issues raised about definition of the devices.**

| Code | Description | Example quote | Number of sites |
|---|---|---|---|
| Medical devices | Refers to issues regarding the question if the remote monitoring technologies used in the study should be defined as medical devices. | *'The researchers indicate that this study does not include medical devices because the technological devices in this study are not used for diagnosis, treatment, or alleviation of disease symptoms. However, the committee needs to be further convinced of this and therefore the committee has some questions about the technological devices.'* | 3 |

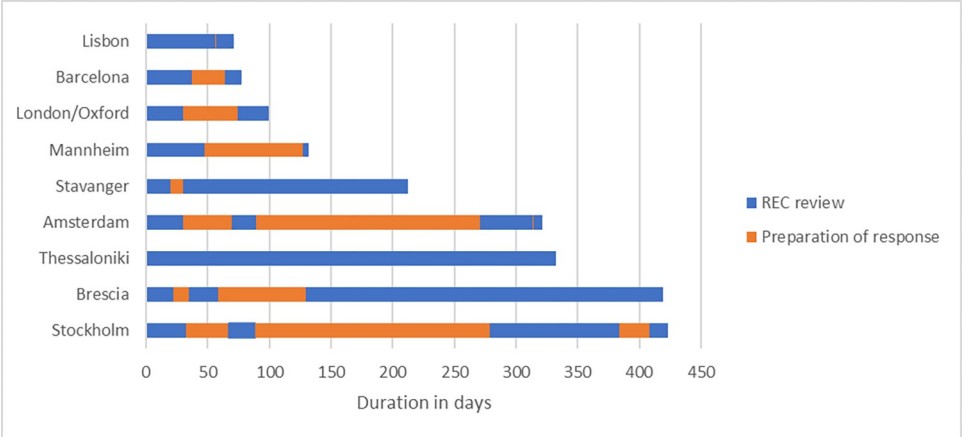

**Fig 1. Duration in days of the REC process per site.** Blue blocks indicate the REC review, while orange blocks indicate the preparation of the response. The long preparation of response time in Amsterdam (second orange block) was because the REC requested approval from the IT department after the second REC review round, which resulted in several months waiting time. More detailed information can be found in the S1 Table.

Centro San Giovanni di Dio Fatebenefratelli (Brescia site) decided to submit the protocol without the wearable camera in the first place. The other sites submitted the protocol with the wearable camera, but based on the REC comments, Zentralinstitut für Seelische Gesundheit Mannheim (Mannheim site), Ace Alzheimer Center Barcelona (Barcelona site), and Karolinska Institutet (Stockholm site) decided to omit the wearable camera from the protocol: *'Uninvolved passers-by/shop visitors/staff etc. are permanently recorded every 30 seconds in public space, without them being able to be informed in time about the circumstance of the recording, e.g. in order to avoid it. After all, the people concerned are not simply photographed, but stored in a database for a research project. [. . .] In any case, the personal rights of those concerned [. . .] are considerably affected by recordings.'* In the other sites, the use of the wearable camera was allowed.

Another set of issues raised by RECs focused on the **participant's wellbeing** (Table 5). As research participants were actively involved in the study for 56 days, RECs raised concerns about several issues related to ensuring that they were comfortable and safe and that their privacy was preserved. Another concern was participant's competence and capacity: most of the comments were related to participants being mentally capable providing informed consent, considering that the study was carried out in a population of dementia patients, for example: '*The Ethics Committee points out that the study may only be carried out in patients who are capable of giving consent. As soon as there are doubts about the capacity to consent, care must be suggested if necessary.'* The REC of Karolinska Institutet (Stockholm site) initially rejected the study protocol because of proportionality: '*The potential benefits for future individuals with early dementia appear unclear. As the risks (significant breach of privacy) for the participants are thus not outweighed by the benefit, the application is rejected'*. To obtain approval, major changes were made to the protocol, i.e., the wearable camera and several questionnaires and clinical tests were omitted from the protocol.

**Methodological issues** concerning the design of the study and the responsibilities of each partner were also frequently addressed. Three sub-themes were found: methodological concerns, recruitment, and liability (Table 6). Particularly on recruitment, major differences between study sites were identified, for example: '*Please provide a clear recruitment process of the three groups of participants, explaining how they will be identified, approached and*

*consented into the study.'* These differences backtracked mainly to the discussion of biological versus clinical definition of AD, i.e., the use of the pathology marker assessments in symptom free individuals. As a result of these discussions, Faculdade de Medicina da Universidade de Lisboa (Lisbon site), and Zentralinstitut für Seelische Gesundheit Mannheim (Mannheim site) decided not to recruit pre-symptomatic individuals into the study.

Finally, RECs raised concerns regarding the **definition of the devices** used in the study. Initially, the protocol was submitted to each REC with the conviction that the RMTs were no medical devices. However, whether the used RMTs should be classified as medical devices (Table 7) was a major issue in Amsterdam UMC (Amsterdam). Medical devices are products or equipment intended for a medical purpose, for example for diagnosis, prevention, monitoring, prediction, prognosis, treatment or alleviation of a disease (European Union Medical Device Regulation (EU no 2017/745), MDR). If devices are classified as medical devices, researchers need to follow specific rules for the submission, assessment and conduct of clinical research using the devices, which are described in the MDR. In the end, the Amsterdam REC was convinced that the RMTs were medical devices: '*The committee is of the opinion that these devices are medical devices, because the devices are used for scientific research with patients with or suspected of having AD, in order to determine the clinical and technical performances and the acceptance of the various devices and thereby the association of the outcome measures with clinical measures for AD.'*. All other sites did not mention the MDR or considered the RMTs as no medical devices after review.

## Discussion

The main goal of this study was to identify the views and concerns of multiple local RECs across Europe and what gaps remain regarding the use of RMTs in clinical research on Alzheimer's disease through a case study of the RADAR-AD project. The main findings were that on one hand generic concerns were raised around data, participants, methodology, and medical devices, while on the other hand the specific focus of concerns differed significantly across study sites. These differences have led to major variance regarding the duration of the approval process, as well as the way the study protocol was implemented. Below, we highlight the major differences and discuss practical considerations regarding the REC approval process on RMT related study protocols.

### Themes

The issues that emerged in the RADAR-AD research ethics process were as expected based on previous literature [30] and can be categorized according to the distinction made by Ienca, Vayena [30] referring to the governance of big data health research. In fact, issues around data fall in their description of "technical" issues, issues about participant's wellbeing are "social issues", issues about study set up are "methodological", and concerns around definition of the devices are "regulatory" in nature. We did not single out ethical issues as each one of these categories were imbued by ethical principles, but privacy was a frequent concern, together with a worry about proportionality of the benefits of the study against the burden imposed on participants.

### Lack of harmonization

More interestingly were the issues that emerged in the comparison of the different processes and the differences between the RECs. As our data was gathered from RECs in different institutions and different European countries, some of the diversity in the responses is due to the lack of harmonization around ethical standards to assess studies in this remit.

The lack of harmonization around evaluative standards can be highly dysfunctional for an international multi-site project. A point in this case is the original rejection of the project by the REC of the Karolinska Institutet (Stockholm site), that resulted in a 1.5 years long approval process when the protocol was finally approved with major changes to the original one. This meant that the data collected by this site was less comparable to the data collected by other sites.

The issue of harmonization is key in health research fields, such as AD, where international multi-site projects are prioritized by funders. Although there are good reasons why some local differences are kept to address cultural differences, harmonizing the process is an important requirement as it saves resources for large research consortia and assures protocol consistency and a sharing of best practices. This lack of harmonization was already found in 2005 in a study involving dementia patients [31], and appears to be unchanged since then.

For example, we saw that in some cases (e.g., Amsterdam UMC), the REC was not the only governance body in charge of assessing the study, and the DPO was also involved in the process. Considering the importance of harmonized interpretation of the General Data Protection Regulation (GDPR), a key area of expertise of a DPO, we found that the involvement of DPO in the approval process of study protocols that involve RMT measurements was highly beneficial. DPOs are key in assessing the data vulnerability of remote assessments, and can provide researchers with important insights on this topic which may not necessarily be provided by an REC. Based on this experience, we suggest that DPOs are involved early in the design of RMT based projects, preferably before the submission of the protocol to the REC, which might even shorten approval duration. Interesting to notify though, was that the interpretation of the DPOs consulted for this study was not similar for all sites. For example, the wearable camera was not approved in the site in Mannheim, after consultation of the DPO, while the camera was approved in the site in Amsterdam, after consultation of the DPO, although both sites are in the European Union and should therefore both adhere to the GDPR.

Another example of lack of harmonization, within the context of AD, is an important difference between countries and institutions concerning the use of biomarker data in asymptomatic individuals. This point concerns rather the view of Alzheimer's disease in general instead of issues with RMTs, but it is still an important example of how the view of RECs differs between countries. Within RADAR-AD, several RECs raised serious concerns regarding the inclusion of a so-called 'preclinical AD' group, i.e., individuals without cognitive complaints but with positive AD biomarkers. The preclinical AD group raises the issue of defining healthy and cognitively normal participants as 'having AD', while there is no treatment or certainty that these participants will develop cognitive impairment in the future. Preclinical AD is a widely accepted study group in research [32], but is not used in clinical practice, due to this lack of treatment and uncertainty of progression. In our view, RECs refusing to include preclinical AD in a clinical study are therefore withholding from an important step forward in clinical trials for AD. This issue is not likely to be reconciled easily, as this relates to epistemic and normative differences in the biological versus clinical view of AD as a disease entity [32]. We do recommend that these scientific differences are acknowledged and discussed by researchers in their applications to enable RECs to understand the rationale behind the study design.

The last issue that would benefit from harmonization is the interpretation of the definition of the used RMTs. If RMTs are classified as medical devices, they should apply to the European MDR, which requires more paperwork and rules to adhere to. Although all participating sites in this study are in the European Union (except the site in Geneva, Switzerland), and thus should adhere to the same rules and regulations regarding GDPR and MDR, the RECs decided differently on the definition of the RMTs: in the Amsterdam site, the RMTs were considered

medical devices, while the other sites considered the RMTs as no medical devices. On the one hand, the RMTs are used to monitor disease symptoms, and can therefore be considered medical devices, but on the other hand, the RMTs in RADAR-AD are used for research purposes only and will not be used to diagnose, treat, or alleviate disease symptoms. The Clinical Trial Information System (CTIS), starting January 31st, 2023, is a central portal from the European Medicines Agency (EMA) to coordinate the submission of drug trials on a European level, which might help in the harmonization of REC decisions within a multi-site study, at least within a study.

## Involvement of patient advisory board

RECs and DPOs should not be the only bodies assessing research using mobile sensors on vulnerable populations. As suggested by Breslin et al. [15], the specific groups and communities who are part of the research should be involved in discussing and addressing the ethical issues and concerns. This is of particular importance in the case of vulnerable groups and for some conditions, such as dementia, where preconceptions and stereotypes about people living with the disease still prevail. The RADAR-AD project set up a Patient Advisory Board (PAB), composed of people with mild cognitive impairment, people with dementia and caregivers, which was operational from the very start of the project and provided valuable input throughout the whole research process. For example, the PAB provided input on the study protocol, participant-facing documents, device selection [33], recruitment, challenges encountered during the COVID pandemic and various ethical issues. The new clinical trials regulation (Regulation (EU) No 536/2014), requires RECs to include a patient representative, which is a start, but remains difficult with dementia patients. Although the PAB was not directly involved in the process of obtaining ethical approval from the RECs, some of the concerns and issues raised by members of the PAB resonated with those raised by the RECs. However, many of these issues and concerns were discussed in a more nuanced way, often seeking a balance between respect for autonomy and non-maleficence, and reflecting on issues such as personhood, stigma, trust, and equity, though it is noteworthy that concerns about aptitude and capacity to use RMTs was not of concern. This led to ideas to address specific concerns and thereby support future participants. In relation to the consent of participants with dementia, for example, the PAB emphasized the importance of supported decision making and a dynamic ongoing consent process as opposed to a more rigid, one-stop consent procedure. Similarly, they proposed the wearable camera being optional, and to provide participants with a card explaining how people's images would be protected and with details of the study in case anyone being photographed asked. With regard to study partners, they also questioned the need for healthy control participants to have them. In addition, they challenged the assumption that all participants with mild cognitive impairment or dementia would need one, accepting nevertheless a principle of equality but with concerns about discrimination against people who live alone, further suggesting the need to consider alternatives such as volunteer study partners.

The differences between the PAB and REC perspectives and concerns are not surprising given that RECs are almost entirely composed of professionals who tend to have a more precautionary approach. These differences raise questions about how to balance the enthusiasm, dedication and problem-solving approach of people with the condition (i.e., to make research better by sharing their experience and perspectives with researchers, focusing on what is still possible and exercising their right to voice) with the more protection/liability-based approach of many RECs. The differences also emphasize the need for greater exchange and communication between lay- and expert governance structures in medical research, particularly in research using remote monitoring technologies where the active involvement of people

affected by the condition in the study design is central. How such exchange and communication can be implemented in context where patients' expertise is valued alongside professional assessment of potential vulnerabilities and risks for participants should be the focus of further research.

## Internal ethics during the study

Another aspect to consider is that although the study methodology at large has been questioned by some REC reviews (e.g. Amsterdam UMC), none of the REC reviews received by RADAR-AD sites questioned the ethics support within the project. Research projects often have a dedicated space for ethics research that is expected to strengthen the legitimacy and social desirability of the project, however the internal ethics structure is not considered as compulsory information for REC applications and, in the case of the RADAR-AD project was not questioned. However, experience has shown that ethics support should be an integral part throughout a project rather than a one-off consideration when seeking ethical approval [34]. The medical ethics model, where ethics committees are called upon to discuss complex cases is relevant here. There have been many studies and experiments in these areas [35], but this is not a requirement by funding organizations or by research governance bodies. Especially in a context where innovative technologies are used that disrupt the traditional way of conducting research, it is important to further develop feasible and sustainable ways of building ethical support into the project.

## Limitations

An important limitation for the current study is that the data used, i.e., communications with RECs and DPOs on the RADAR-AD study protocol, was not originally collected as such. Neither did the consortium plan to use these communications for study purposes. We therefore did not collect additional data such as interviews with REC members to better understand their points and considerations. Future research should include interviews or focus groups with REC members to better understand their views on the topics discussed in this discussion, i.e., the involvement of the DPO and PAB, interpretation of the MDR, and internal ethics. At the same time, the material already offered an interesting insight into REC procedures and concerns towards these types of projects.

## Conclusion

To conclude, this study highlights the generic concerns raised by RECs regarding data, participants, methodology, and governance of clinical research protocols using RMTs, while important differences in the view of RECs remain present. As these differences may have important practical consequences, which could lead to significant delays in the approval process, we highly recommend a further harmonization on and further research to specific elements of the approval process. This includes, but is not limited to, the involvement of data protection officers, patient advisory boards, and ethics support. As these are complicated matters, routes towards harmonization should be initiated on a national or even European level.

## Supporting information

**S1 Table. Details of the documents reviewed for this study and review processes in each site.**
(DOCX)

## Acknowledgments

We would like to acknowledge and thank the members of the RADAR-AD Patient Advisory Board for their input to the work described in this article. We thank all past and present RADAR-AD consortium members for their contribution to the project (in alphabetical order): Dag Aarsland, Halil Agin, Vasilis Alepopoulos, Alankar Atreya, Sudipta Bhattacharya, Virginie Biou, Joris Borgdorff, Anna-Katharine Brem, Neva Coello, Pauline Conde, Nick Cummins, Jelena Curcic, Casper de Boer, Yoanna de Geus, Paul de Vries, Ana Diaz, Richard Dobson, Aidan Doherty, Andre Durudas, Gul Erdemli, Amos Folarin, Suzanne Foy, Holger Froehlich, Jean Georges, Dianne Gove, Margarita Grammatikopoulou, Kristin Hannesdottir, Robbert Harms, Mohammad Hattab, Keyvan Hedayati, Chris Hinds, Adam Huffman, Dzmitry Kaliukhovich, Irene Kanter-Schlifke, Ivan Koychev, Rouba Kozak, Julia Kurps, Sajini Kuruppu, Claire Lancaster, Robert Latzman, Ioulietta Lazarou, Manuel Lentzen, Federica Lucivero, Florencia Lulita, Nivethika Mahasivam, Nikolay Manyakov, Emilio Merlo Pich, Peyman Mohtashami, Marijn Muurling, Vaibhav Narayan, Vera Nies, Spiros Nikolopoulos, Andrew Owens, Marjon Pasmooij, Dorota Religa, Gaetano Scebba, Emilia Schwertner, Rohini Sen, Niraj Shanbhag, Laura Smith, Meemansa Sood, Thanos Stavropoulos, Pieter Stolk, Ioannis Tarnanas, Srinivasan Vairavan, Nick van Damme, Natasja van Velthogen, Herman Verheij, Pieter Jelle Visser, Bert Wagner, Gayle Wittenberg, and Yuhao Wu.

https://www.radar-ad.org/

## Author Contributions

**Conceptualization:** Marijn Muurling, Federica Lucivero, Casper de Boer.

**Data curation:** Marijn Muurling, Federica Lucivero, Casper de Boer.

**Formal analysis:** Marijn Muurling, Federica Lucivero, Casper de Boer.

**Funding acquisition:** Pieter Jelle Visser, Dag Aarsland.

**Investigation:** Marijn Muurling, Ivan Koychev, Dora Roik, Lutz Froelich, Emilia Schwertner, Dorota Religa, Carla Abdelnour, Mercè Boada, Monica Almici, Samantha Galluzzi, Sandra Cardoso, Alexandre de Mendonça, Andrew P. Owens, Sajini Kuruppu, Martha Therese Gjestsen, Ioulietta Lazarou, Mara Gkioka, Magda Tsolaki, Federica Lucivero, Casper de Boer.

**Methodology:** Marijn Muurling, Federica Lucivero, Casper de Boer.

**Project administration:** Marijn Muurling, Pieter Jelle Visser, Dag Aarsland, Casper de Boer.

**Resources:** Marijn Muurling, Ivan Koychev, Dora Roik, Lutz Froelich, Emilia Schwertner, Dorota Religa, Carla Abdelnour, Mercè Boada, Monica Almici, Samantha Galluzzi, Sandra Cardoso, Alexandre de Mendonça, Andrew P. Owens, Sajini Kuruppu, Martha Therese Gjestsen, Ioulietta Lazarou, Mara Gkioka, Magda Tsolaki, Casper de Boer.

**Supervision:** Pieter Jelle Visser, Dag Aarsland, Casper de Boer.

**Visualization:** Marijn Muurling, Federica Lucivero, Casper de Boer.

**Writing – original draft:** Marijn Muurling, Anna M. G. Pasmooij, Federica Lucivero, Casper de Boer.

**Writing – review & editing:** Marijn Muurling, Anna M. G. Pasmooij, Ivan Koychev, Dora Roik, Lutz Froelich, Emilia Schwertner, Dorota Religa, Carla Abdelnour, Mercè Boada, Monica Almici, Samantha Galluzzi, Sandra Cardoso, Alexandre de Mendonça, Andrew P. Owens, Sajini Kuruppu, Martha Therese Gjestsen, Ioulietta Lazarou, Mara Gkioka, Magda

Tsolaki, Ana Diaz, Dianne Gove, Pieter Jelle Visser, Dag Aarsland, Federica Lucivero, Casper de Boer.

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
