## [Decision Letter · Decision Letter 0]

27 Mar 2023

PONE-D-23-05719Ethical challenges of using remote monitoring technologies for clinical research: a case study of the role of local Research Ethics Committees in the RADAR-AD studyPLOS ONE

Dear Dr. Muurling,

Thank you for submitting your manuscript to PLOS ONE. After careful consideration, we feel that it has merit but does not fully meet PLOS ONE’s publication criteria as it currently stands. Therefore, we invite you to submit a revised version of the manuscript that addresses the points raised during the review process.

We look forward to receiving your revised manuscript.

Kind regards,

Dinh-Toi Chu, PhD

Academic Editor

PLOS ONE

Journal Requirements: 

Review Comments to the Author

Reviewer #1: The study employs qualitative analysis to discover commonalities and differences in how research ethics committees (RECs) evaluated a large-scale multinational research project on the use of remote monitoring technologies (RMTs) for people suffering from Alzheimer’s disease. It compares ethics reviews from RECs in 9 European countries in terms of issues raised, process duration, and the involvement of a data protection officer.

This is an excellent manuscript. Particularly impressive is the authors’ ability to compare the evaluation of the same project across 9 countries. While the perspective of RECs on RMTs is interesting, the value of the manuscript is elevated by the discussion of the lack of harmonization of REC processes. This topic will only gain greater prominence as the harmonization of research processes across the EU picks up and the study will add a valuable contribution to coming debates.

Aims and scope of the study are well-described and motivated. The study context is explained well. The background information provided on remote monitoring technologies and the role of RECs is clear and sufficient; connected ethical issues are described succinctly.

The description of the qualitative methodology misses a reference to the coding approach used and could be slightly more detailed. It is unclear to what degree, if any, peer-coding was used and whether all authors involved in coding were coding all of the material or shared the workload between them. I also missed an overview over which REC provided which types of documents for the analysis (this is not crucial for the manuscript and could be included as supplementary material).

A more detailed overview table over the differing review processes at each site with additional information on intermediary steps might also be helpful and would help strengthen the study’s contribution to REC harmonization efforts (Fig. 1 currently only shows process duration in a coarse manner). I would also be interested in learning about the level of harmonization of REC processes in each country: i.e. to what degree is the process site-specific or country-specific?

It might also help, but is probably out of scope for this study, to relate the findings to country-specific cultural attitudes, e.g. relating to data privacy.

The policy recommendations of the authors are sound and well-argued, but the structure of the discussion section is currently somewhat unclear and could benefit from either being broken up into smaller sections or from additional signposting at the beginning of the section. “Experts by learning” (326) is missing a reference if this is intended to be a quote.

Reviewer #2: The manuscript under review describes the challenges of ethical approval for a large European Study on remote monitoring for Alzheimer’s disease. The research ethics review process at 10 sites in 9 European countries were reported and compared. It is a relevant study that illustrates that ethics review remains a human activity where evaluation of risks between research ethics committees might differ, and the consequences this might have for research.

Although the authors write that written permission was obtained from the REC to use direct anonymized quotes for publication, I question whether RECs have given their permission that they are named in the article and that specific information about their RECs evaluation (e.g. duration, position on medical device, code frequencies) is being provided. In my view, illustrating the challenges of the current governance structures is possible without naming these committees.

With regard to the duration of the process of ethics approval, the current figures are not sufficiently correct to provide a clear representation of the process. A process of an ethics approval consists of a process between two parties, with questions and answers. So, in order to provide an honest representation it is necessary to detail how long the REC delayed an answer and how long the researchers delayed their answer to the questions of the REC. This is necessary in order to provide a fair representation. If this is not possible, this should be removed.

Although the current analysis is clear with regard to several ethical aspects of research, I was wondering whether interpretations of DPOs with regard to data processing (e.g. legal grounds for processing data) were similar in all sites.

On lines 300-302, the authors refer to the specific groups and communities that should be part of the discussion. The clinical trial regulations obliges now the presence of patient representatives. It might be a relevant addition, even if not every patient group can be represented in an ethics committee.

On line 301, please add the authors’ name.

On line 330, please explain anachronistic.

Reviewer #3: This is an interesting paper that lends support to a new era of digitizing medical research, namely the inclusion of "wearables" and remote monitoring technologies (RMTs). However, there is a significant disconnect between what the authors propose to study and then what is discussed after reporting the results. The results demonstrate a series of thematic analyses that present terms most frequently appearing in the research ethics committee review of the RADAR-AD study across the different study sites. Yet, this qualitative study discussed very few themes qualitatively in the discussion, instead beginning new threads of discussion about the RADAR-AD patient advisory board and specific definitions of RADAR-AD study population, a critique that undermines the authors' central thesis. Lastly, there is no promising conclusion or direction for future research with these qualitative findings in mind. Another discussion point that stood out to me was labeling the Swedish REC as "too strict" when earlier in the paper the themes identified from the initial rejection of the study protocol were legitimate concerns. The IT, data management and several other data and research governance safeguards the Sweden REC requested could be argued in the opposing direction. For instance, maybe the other countries were too quick to approve a study with a novel RMT and ill-defined data privacy and safety, putting vulnerable study participants at risk for a study that the authors critiqued as presenting epistemological flaws (e.g., preclinical AD participants recruited without concrete value to the study).

To help get started on the major revision, I will present the following revisions:

*Lines 95 - 96: RECs ENFORCE research governance through a review mechanism, reviewing study protocol, methods, informed consent forms and data safety, management and monitoring plans. I see the idea of research governance used in different contexts in the paper - RECs enforce research governance, which is an all encompassing term applicable to everything and everyone involved in clinical research.

*Line 97-98: Not usually. For an REC to have quorum, stakeholders who bring a diverse set of perspectives MUST be present (e.g., patient advocates, legal experts, medical professionals, ethicists, community, etc.)

*Lines 98-99: "An issue affecting multi-site studies is that research governance processes vary substantially across institutions and countries". This is the heart of the paper, which studies the multi-site RADAR-AD trial and the variation in how research governance is enforced through respective RECs. This sentence should be more clearly worded. Try: "Variation in how research governance practices are enforced across different institutions and countries participating in multi-site studies is a concern".

*Check line 100-101 for concise + grammar. logical cohesion in writing

*Line 102: Reword to "We must better understand the "needs, views and attitudes" of REC members wherever health-related RMTs are used as a source of "big data" in research."

*Line 103: delete "aims to"

*Line 105: This paper also explores...remove "intends to"

*Line 106: REC, committees -- repetitive

@Editorial Team - Table headings vary in formatting, e.g., Table 2 is italicized but Table 1 is not.

*Line 181-182: This does not hold true according to data presented in Table 3. Lisbon and Thessaloniki sites report 0% code frequencies for data-related terms.

*Line 285-286: My understanding of this paper is that the RADAR-AD study is used as a case study of how RMTs are viewed and handled by different RECs within the same multisite study - the structures that enforce research governance. Diving into an a critique of how RADAR-AD defines its study population is a counter example to your argument - it instead supports the idea that the RADAR-AD study was conceptually flawed to begin with and because of a methodologically-weak study design, the risks of data privacy posed by RMTs greatly outweighed any benefit of the research. This is a separate concern from how different RECs handle RMTs embedded in clinical trials, even the same clinical trial.

*Line 291 and beyond: Please expand on this discussion and make this more clear in the results section for emphasis. How exactly were RMTs classified in the study overall, at each study site, and then how is an RMT supposed to be classified according to the European MDR and for each country's regulations. Is there a certain feature / quality that makes an RMT a medical device? For instance, this is not intuitive. a contact lens worn in place of glasses is a medical device although we use it daily and it seems low risk. Is there a feature of a Fitbit or wrist-worn tracker that makes it a medical device or is the tracking of data (e.g., heartrate, pulse, respiratory rate) in a research study, is what implies that it should be treated as a medical device?

*Line 297 - if CTIS will help coordinate the submission of drug trials - how will this help harmonize how medical devices are defined by RECs and trial investigators? Drug and device trials differ very much, especially since device trials often present far more lenient regulatory and ethical oversight compared to drug trials as they are viewed as less invasive.

*Line 301, please identify the groups or authors instead of adding the citation

*Line 326: Please cite the quotation or advise why this is quoted.

*Line 362: Study results are reported in the discussion as "the material was very rich". This does not add value and should not be included.

Please also include a description of the type of study the RADAR-AD study was. Was this a pragmatic, late phase trial? Did this study aim to collect real world information? What was the purpose of the study and what benefit to society redeemed the risks of data privacy, safety and well-being posed to participants? Lastly, there is no discussion of DSMB (data safety monitoring boards) that can continue to evaluate the safety of the data and report any risks perceived during the study and then ask the study sites to intervene for the safety of participants. This is also an additional safeguard that can reinforce shorter review times at the REC stage.

A final note is a lack of connection to existing literature. There are many studies on data governance, data privacy, big data, health records and the participation/consent of vulnerable communities like those living with AD and dementia. Please see the work of Largent et al like the paper "Ethical and Regulatory Issues for Embedded Pragmatic Trials Involving People Living With Dementia" and Clinical News: Patient Data Safety for Dementia Patients Using Apps by Lisa Rosenfeld, MD, MPH, and Ipsit Vahia, MD (https://www.todaysgeriatricmedicine.com/archive/JF18p8.shtml).

Thank you for your work on this VERY interesting and complex topic.

---

## [Author Response · Author response to Decision Letter 0]

4 Apr 2023

Dear editor,

We would like to thank reviewer 1, 2 and 3 for their constructive comments on our manuscript entitled ‘Ethical challenges of using remote monitoring technologies for clinical research: a case study of the role of local Research Ethics Committees in the RADAR-AD study’. The comments were relevant and enabled us to improve the quality of our manuscript. Below, please find a point by point response to the provided comments:

Reviewer #1: The study employs qualitative analysis to discover commonalities and differences in how research ethics committees (RECs) evaluated a large-scale multinational research project on the use of remote monitoring technologies (RMTs) for people suffering from Alzheimer’s disease. It compares ethics reviews from RECs in 9 European countries in terms of issues raised, process duration, and the involvement of a data protection officer.

This is an excellent manuscript. Particularly impressive is the authors’ ability to compare the evaluation of the same project across 9 countries. While the perspective of RECs on RMTs is interesting, the value of the manuscript is elevated by the discussion of the lack of harmonization of REC processes. This topic will only gain greater prominence as the harmonization of research processes across the EU picks up and the study will add a valuable contribution to coming debates.

Aims and scope of the study are well-described and motivated. The study context is explained well. The background information provided on remote monitoring technologies and the role of RECs is clear and sufficient; connected ethical issues are described succinctly.

1. The description of the qualitative methodology misses a reference to the coding approach used and could be slightly more detailed. It is unclear to what degree, if any, peer-coding was used and whether all authors involved in coding were coding all of the material or shared the workload between them. 

References to the coding approach are added in lines 146-150. The authors shared the workload between them for the first coding round, but after that, MM reviewed all coded documents again to harmonize the coding. This is added in lines 151-152 of the manuscript.

2. I also missed an overview over which REC provided which types of documents for the analysis (this is not crucial for the manuscript and could be included as supplementary material).

An overview over which documents were reviewed for this study is added in a table in the supplementary material. 

3. A more detailed overview table over the differing review processes at each site with additional information on intermediary steps might also be helpful and would help strengthen the study’s contribution to REC harmonization efforts (Fig. 1 currently only shows process duration in a coarse manner). 

A more detailed overview of the review process at each site is added to as an additional column to the table in the supplementary material. Please see the last column of the table as discussed in the previous question.

4. I would also be interested in learning about the level of harmonization of REC processes in each country: i.e. to what degree is the process site-specific or country-specific?

We added a new column in Table 2 to describe if the RECs operate on a site-specific, region-specific, or country-specific level.

5. It might also help, but is probably out of scope for this study, to relate the findings to country-specific cultural attitudes, e.g. relating to data privacy.

We agree with the reviewer that country specific cultural attitudes of our interest. However, to our opinion, this topic in itself it too large to be included in this study. Cultural attitudes towards dementia and use of technology are highly dependent on many factors including regional/institutional attitudes, state of digital development of a country/region, and types of medical professions included in the clinical center involved. To this end, we decided not to include a discussion of this topic in order to do justice to the breath of this potential discussion.

6. The policy recommendations of the authors are sound and well-argued, but the structure of the discussion section is currently somewhat unclear and could benefit from either being broken up into smaller sections or from additional signposting at the beginning of the section. 

We have added sub-headers and added signposting at the beginning of each section in the discussion section.

7. “Experts by learning” (326) is missing a reference if this is intended to be a quote.

These words were not intended to be quoted. We have changed it to ‘professionals’ to avoid confusion (line 359).

Reviewer #2: The manuscript under review describes the challenges of ethical approval for a large European Study on remote monitoring for Alzheimer’s disease. The research ethics review process at 10 sites in 9 European countries were reported and compared. It is a relevant study that illustrates that ethics review remains a human activity where evaluation of risks between research ethics committees might differ, and the consequences this might have for research.

1. Although the authors write that written permission was obtained from the REC to use direct anonymized quotes for publication, I question whether RECs have given their permission that they are named in the article and that specific information about their RECs evaluation (e.g. duration, position on medical device, code frequencies) is being provided. In my view, illustrating the challenges of the current governance structures is possible without naming these committees.

We specifically asked each REC if their name could be included in the publication as well. To our opinion, it is of additional value to name the specific RECs, as the challenges in governance structure could be dependent on regional or institutional attitudes. Also, the RECs involved in the RADAR-AD study are also named in other public documents, such as project deliverables.

2. With regard to the duration of the process of ethics approval, the current figures are not sufficiently correct to provide a clear representation of the process. A process of an ethics approval consists of a process between two parties, with questions and answers. So, in order to provide an honest representation it is necessary to detail how long the REC delayed an answer and how long the researchers delayed their answer to the questions of the REC. This is necessary in order to provide a fair representation. If this is not possible, this should be removed.

We have added a table in the supplementary material which elaborates more on the review process per site and which bodies were involved in each round.

3. Although the current analysis is clear with regard to several ethical aspects of research, I was wondering whether interpretations of DPOs with regard to data processing (e.g. legal grounds for processing data) were similar in all sites.

The interpretations of DPOs were not similar for all sites. For example, the use of the wearable camera was not approved in Germany after a consult with a DPO, while the camera was approved in the Netherlands after a consult with a DPO. This was added in the discussion (lines 301-305).

4. On lines 300-302, the authors refer to the specific groups and communities that should be part of the discussion. The clinical trial regulations obliges now the presence of patient representatives. It might be a relevant addition, even if not every patient group can be represented in an ethics committee.

A sentence about the clinical trial regulation has been added in lines 340-342.

5. On line 301, please add the authors’ name.

The authors’ name is added.

6. On line 330, please explain anachronistic.

We have removed the word anachronistic and simplified the sentence (line 362-363).

Reviewer #3: This is an interesting paper that lends support to a new era of digitizing medical research, namely the inclusion of "wearables" and remote monitoring technologies (RMTs). 

1. However, there is a significant disconnect between what the authors propose to study and then what is discussed after reporting the results. The results demonstrate a series of thematic analyses that present terms most frequently appearing in the research ethics committee review of the RADAR-AD study across the different study sites. Yet, this qualitative study discussed very few themes qualitatively in the discussion, instead beginning new threads of discussion about the RADAR-AD patient advisory board and specific definitions of RADAR-AD study population, a critique that undermines the authors' central thesis. 

We agree that the issued found in the results section were not discussed elaborately in the discussion. The rationale for this was that the results were expected and did not report new discoveries. The gaps, or themes that were not mentioned were more interesting, and were therefore discussed. To identify the gaps that remain after the REC review was one of the aims of the study. We have added the sentences in lines 259, 268-269 and 278-279 to clarify this more. 

2. Lastly, there is no promising conclusion or direction for future research with these qualitative findings in mind. 

We have added suggestions for future research including interviews and focus groups with REC members to better understand their views (lines 390-393 and 401), and future research how to involve patients (lines 366-369). 

3. Another discussion point that stood out to me was labeling the Swedish REC as "too strict" when earlier in the paper the themes identified from the initial rejection of the study protocol were legitimate concerns. The IT, data management and several other data and research governance safeguards the Sweden REC requested could be argued in the opposing direction. For instance, maybe the other countries were too quick to approve a study with a novel RMT and ill-defined data privacy and safety, putting vulnerable study participants at risk for a study that the authors critiqued as presenting epistemological flaws (e.g., preclinical AD participants recruited without concrete value to the study). 

We acknowledge that the reasoning could be argued in the opposing way. The exact (translated) words of the Swedish REC to reject the study were: 

The Ethics Review Authority notes that the research project concerns a group which, due to its disease symptoms (memory disorder due to Alzheimer's disease), may be considered a particularly vulnerable group who may be expected to have difficulty taking an active part in participating in the study. The participants are exposed to a significant invasion of privacy and have no personal benefit from participating in the study. The potential benefits for future individuals with early dementia appear unclear. As the risks (significant breach of privacy) for the participants are thus not outweighed by the benefit, the application is rejected. 

This shows that their concern was specifically focused on the dementia group, and not on the preclinical AD group. Including a preclinical AD group in research is widely accepted in the AD field (1), especially when the 

biomarker status is not disclosed to participants. The argument ‘may be expected to have difficulty’, is not very strong, as RADAR-AD was specifically designed to discover if this dementia group was able to use these kinds of RMTs. Moreover, we would like to point out that the RMTs used in this study are not novel devices or apps. We used RMTs that were already developed, either for research purposes or commercially available. We have added this clarification in lines 118-120. At last, the data privacy and safety of our RMTs was well defined, as we had two work packages within the project specifically appointed to build a secure data platform and focus on research ethics and safety. Moreover, the data platform and RMTs were checked on safety and privacy by a specialized IT department and data privacy officer in Amsterdam (the sponsor of the study). We, however, acknowledge that the wording of being ‘too strict’ might be not appropriate, and we have therefore changed the wording (lines 283-286).

To help get started on the major revision, I will present the following revisions:

4. Lines 95 - 96: RECs ENFORCE research governance through a review mechanism, reviewing study protocol, methods, informed consent forms and data safety, management and monitoring plans. I see the idea of research governance used in different contexts in the paper - RECs enforce research governance, which is an all-encompassing term applicable to everything and everyone involved in clinical research.

We have changed the sentence in lines 94-97, so that it is clear that RECs enforce research governance.

5. Line 97-98: Not usually. For an REC to have quorum, stakeholders who bring a diverse set of perspectives MUST be present (e.g., patient advocates, legal experts, medical professionals, ethicists, community, etc.)

The word usually is removed from the sentence in lines 97-98.

6. Lines 98-99: "An issue affecting multi-site studies is that research governance processes vary substantially across institutions and countries". This is the heart of the paper, which studies the multi-site RADAR-AD trial and the variation in how research governance is enforced through respective RECs. This sentence should be more clearly worded. Try: "Variation in how research governance practices are enforced across different institutions and countries participating in multi-site studies is a concern".

The sentence is changed to the proposed sentence.

7. Check line 100-101 for concise + grammar. logical cohesion in writing

The sentence has been changed to improve the grammar and cohesion.

8. Line 102: Reword to "We must better understand the "needs, views and attitudes" of REC members wherever health-related RMTs are used as a source of "big data" in research."

The sentence is reworded to the proposed sentence.

9. Line 103: delete "aims to"

‘Aims to’ is deleted.

10. Line 105: This paper also explores...remove "intends to"

‘Intends to’ is removed.

11. Line 106: REC, committees – repetitive

Changed to RECs.

12. @Editorial Team - Table headings vary in formatting, e.g., Table 2 is italicized but Table 1 is not.

All table headings are now italicized.

13. Line 181-182: This does not hold true according to data presented in Table 3. Lisbon and Thessaloniki sites report 0% code frequencies for data-related terms.

We added ‘except for the sites in Lisbon and Thessaloniki’. 

14. Line 285-286: My understanding of this paper is that the RADAR-AD study is used as a case study of how RMTs are viewed and handled by different RECs within the same multisite study - the structures that enforce research governance. Diving into an a critique of how RADAR-AD defines its study population is a counter example to your argument - it instead supports the idea that the RADAR-AD study was conceptually flawed to begin with and because of a methodologically-weak study design, the risks of data privacy posed by RMTs greatly outweighed any benefit of the research. This is a separate concern from how different RECs handle RMTs embedded in clinical trials, even the same clinical trial.

We agree that the point discussed in this paragraph does not concern an issue with RMTs, but with the view of Alzheimer’s disease in general. It is however an example of how the view of RECs differs between countries. Importantly, the issues raised by several REC’s did concern the inclusion of a preclinical AD group within the context of RMT measurements, which does link to the goals of this study. Preclinical AD is a widely accepted study group in research, but is not used in clinical practice, as there is no treatment yet. In our view, RECs refusing to include preclinical AD in a clinical study are therefore withholding from an important step forwards in clinical trials for AD, and is therefore an important point to mention in the discussion of this manuscript.

15. Line 291 and beyond: Please expand on this discussion and make this more clear in the results section for emphasis. How exactly were RMTs classified in the study overall, at each study site, and then how is an RMT supposed to be classified according to the European MDR and for each country's regulations. Is there a certain feature / quality that makes an RMT a medical device? For instance, this is not intuitive. a contact lens worn in place of glasses is a medical device although we use it daily and it seems low risk. Is there a feature of a Fitbit or wrist-worn tracker that makes it a medical device or is the tracking of data (e.g., heartrate, pulse, respiratory rate) in a research study, is what implies that it should be treated as a medical device?

The RMTs were classified as medical devices in Amsterdam, but as no medical device (or not mentioned at all) in the other sites. A quote on the rationale for this decision of the REC in Amsterdam was already included in the results section (lines 248-252). We have added a more elaborate explanation in the results section (lines 240-245 and 252-253) and more elaborate discussion in the discussion section (lines 320-324). 

16. Line 297 - if CTIS will help coordinate the submission of drug trials - how will this help harmonize how medical devices are defined by RECs and trial investigators? Drug and device trials differ very much, especially since device trials often present far more lenient regulatory and ethical oversight compared to drug trials as they are viewed as less invasive.

The CTIS will help in harmonizing the decision of different RECs within one multi-site study, rather than the harmonization of medical devices across studies. We have added a clarification in lines 327-328.

17. Line 301, please identify the groups or authors instead of adding the citation

The authors are added.

18. Line 326: Please cite the quotation or advise why this is quoted.

We have changed the quoted words for professionals.

19. Line 362: Study results are reported in the discussion as "the material was very rich". This does not add value and should not be included.

The sentence is reworded.

20. Please also include a description of the type of study the RADAR-AD study was. Was this a pragmatic, late phase trial? Did this study aim to collect real world information? What was the purpose of the study and what benefit to society redeemed the risks of data privacy, safety and well-being posed to participants? Lastly, there is no discussion of DSMB (data safety monitoring boards) that can continue to evaluate the safety of the data and report any risks perceived during the study and then ask the study sites to intervene for the safety of participants. This is also an additional safeguard that can reinforce shorter review times at the REC stage.

The type of study and aims of RADAR-AD are discussed in the materials and methods section (lines 113-121). We added a reference to the project website to get a clearer idea about the project (line 113). The study is already finished and collected real-world data from more than 220 participants, as discussed in lines 115 and 117. There was no DSMB during the study, we added a sentence (lines 120-121) to explain this.

21. A final note is a lack of connection to existing literature. There are many studies on data governance, data privacy, big data, health records and the participation/consent of vulnerable communities like those living with AD and dementia. Please see the work of Largent et al like the paper "Ethical and Regulatory Issues for Embedded Pragmatic Trials Involving People Living With Dementia" and Clinical News: Patient Data Safety for Dementia Patients Using Apps by Lisa Rosenfeld, MD, MPH, and Ipsit Vahia, MD (https://www.todaysgeriatricmedicine.com/archive/JF18p8.shtml).

We have added new references to the introduction, in lines 84 and 87 to connect more to existing literature.

References

1. Jack CR, Jr., Bennett DA, Blennow K, Carrillo MC, Dunn B, Haeberlein SB, et al. NIA-AA Research Framework: Toward a biological definition of Alzheimer's disease. Alzheimers Dement. 2018;14(4):535-62.

---

## [Decision Letter · Decision Letter 1]

19 Apr 2023

PONE-D-23-05719R1Ethical challenges of using remote monitoring technologies for clinical research: a case study of the role of local Research Ethics Committees in the RADAR-AD studyPLOS ONE

Dear Dr. Muurling,

Thank you for submitting your manuscript to PLOS ONE. After careful consideration, we feel that it has merit but does not fully meet PLOS ONE’s publication criteria as it currently stands. Therefore, we invite you to submit a revised version of the manuscript that addresses the points raised during the review process.

We look forward to receiving your revised manuscript.

Kind regards,

Dinh-Toi Chu, PhD

Academic Editor

PLOS ONE

 **********

Review Comments to the Author:

Reviewer #1: I thank the authors for addressing my previous concerns and suggestions. I have noticed 2 minor issues that should be addressed before the manuscript is ready for publication:

1) The table in the supplementary material (line 514) is currently missing the content for the column entitled “documents reviewed”.

2) Table 2 (line 141) needs a brief in-text explanation of what is meant by site-specific, region-specific, and country-specific.

Reviewer #2: (No Response)

Reviewer #3: The revision has now addressed the comments of reviewers and provided explanations for the comments they disagreed with. In the responses, the authors gave a good explanation of the study design's inclusion of pre-clinical AD. They agreed that the study design "does not concern an issue with RMTs, but with the view of Alzheimer’s disease in general. It is however an example of how the view of RECs differs between countries. Importantly, the issues raised by several REC’s did concern the inclusion of a preclinical AD group within the context of RMT measurements, which does link to the goals of this study. Preclinical AD is a widely accepted study group in research, but is not used in clinical practice, as there is no treatment yet. In our view, RECs refusing to include preclinical AD in a clinical study are therefore withholding from an important step forwards in clinical trials for AD, and is therefore an important point to mention in the discussion of this manuscript." I believe this is a great point to include in the actual manuscript, particularly in the study description (lines 115-116) or in more detail in the discussion (308-312).

The list of citations includes a #61 that is blank.

There are slight grammatical and spelling errors throughout. Please proofread. E.g., 'publically' line 119-120.

---

## [Author Response · Author response to Decision Letter 1]

21 Apr 2023

Dear editor,

We would like to thank reviewer 1 and 3 for their additional comments on our manuscript entitled ‘Ethical challenges of using remote monitoring technologies for clinical research: a case study of the role of local Research Ethics Committees in the RADAR-AD study’. The comments were relevant and enabled us to improve the quality of our manuscript. Below, please find a point by point response to the provided comments:

Reviewer #1: I thank the authors for addressing my previous concerns and suggestions. I have noticed 2 minor issues that should be addressed before the manuscript is ready for publication:

1. The table in the supplementary material (line 514) is currently missing the content for the column entitled “documents reviewed”.

We have changed the header of the table and the title of the column to make it more clear what the column content entails. 

2. Table 2 (line 141) needs a brief in-text explanation of what is meant by site-specific, region-specific, and country-specific.

We have added an in-text sentence (line 130-132), and also added a footnote to Table 2 (lines 145-147) to clarify what is meant by site-specific, region-specific, and country-specific.

Reviewer #2: (No Response)

Reviewer #3: The revision has now addressed the comments of reviewers and provided explanations for the comments they disagreed with. 

1. In the responses, the authors gave a good explanation of the study design's inclusion of pre-clinical AD. They agreed that the study design "does not concern an issue with RMTs, but with the view of Alzheimer’s disease in general. It is however an example of how the view of RECs differs between countries. Importantly, the issues raised by several REC’s did concern the inclusion of a preclinical AD group within the context of RMT measurements, which does link to the goals of this study. Preclinical AD is a widely accepted study group in research, but is not used in clinical practice, as there is no treatment yet. In our view, RECs refusing to include preclinical AD in a clinical study are therefore withholding from an important step forwards in clinical trials for AD, and is therefore an important point to mention in the discussion of this manuscript." I believe this is a great point to include in the actual manuscript, particularly in the study description (lines 115-116) or in more detail in the discussion (308-312).

Thank you for this suggestion, we have added the explanation in the discussion (lines 313-314 and lines 319-322).

2. The list of citations includes a #61 that is blank.

Thank you for noticing, this 61 belongs however to the previous citation (#35) as part of the page numbers.

3. There are slight grammatical and spelling errors throughout. Please proofread. E.g., 'publically' line 119-120.

We have changed ‘publically’ to ‘publicly’ in line 119. We have proofread the manuscript and corrected grammatical and spelling errors.

---

## [Editor Report · Decision Letter 2]

2 May 2023

Ethical challenges of using remote monitoring technologies for clinical research: a case study of the role of local Research Ethics Committees in the RADAR-AD study

PONE-D-23-05719R2

Dear Dr. Muurling,

We’re pleased to inform you that your manuscript has been judged scientifically suitable for publication and will be formally accepted for publication once it meets all outstanding technical requirements.

Kind regards,

Dinh-Toi Chu, PhD

Academic Editor

PLOS ONE
---

## [Editor Report · Acceptance letter]

26 Jun 2023

PONE-D-23-05719R2 

Ethical challenges of using remote monitoring technologies for clinical research: a case study of the role of local Research Ethics Committees in the RADAR-AD study 

Dear Dr. Muurling:

I'm pleased to inform you that your manuscript has been deemed suitable for publication in PLOS ONE. Congratulations! Your manuscript is now with our production department. 

Kind regards, 

on behalf of

Dr. Dinh-Toi Chu 

Academic Editor

PLOS ONE